# Sperm migration in the genital tract—*In silico* experiments identify key factors for reproductive success

**Jorin Diemer**[1], **Jens Hahn**[1], **Björn Goldenbogen**[1], **Karin Müller**[2]*, **Edda Klipp**[1]*

**1** Theoretical Biophysics, Humboldt-Universität zu Berlin, Germany, **2** Leibniz Institute for Zoo and Wildlife Research, Berlin, Germany

* mueller@izw-berlin.de (KM); edda.klipp@rz.hu-berlin.de (EK)

**Data Availability Statement:** All relevant data are within the manuscript and its Supporting information files.

## Abstract

Sperm migration in the female genital tract controls sperm selection and, therefore, reproductive success as male gametes are conditioned for fertilization while their number is dramatically reduced. Mechanisms underlying sperm migration are mostly unknown, since *in vivo* investigations are mostly unfeasible for ethical or practical reasons. By presenting a spatio-temporal model of the mammalian female genital tract combined with agent-based description of sperm motion and interaction as well as parameterizing it with bovine data, we offer an alternative possibility for studying sperm migration *in silico*. The model incorporates genital tract geometry as well as biophysical principles of sperm motion observed *in vitro* such as positive rheotaxis and thigmotaxis. This model for sperm migration from vagina to oviducts was successfully tested against *in vivo* data from literature. We found that physical sperm characteristics such as velocity and directional stability as well as sperm-fluid interactions and wall alignment are critical for success, i.e. sperms reaching the oviducts. Therefore, we propose that these identified sperm parameters should be considered in detail for conditioning sperm in artificial selection procedures since the natural processes are normally bypassed in reproductive *in vitro* technologies. The tremendous impact of mucus flow to support sperm accumulation in the oviduct highlights the importance of a species-specific optimum time window for artificial insemination regarding ovulation. Predictions from our extendable *in silico* experimental system will improve assisted reproduction in humans, endangered species, and livestock.

## Author summary

Successful fertilisation is indispensable for all mammals' existence. Positioning of living sperms within the female genital tract and migration towards the oviducts is a prerequisite for fusion of sperm and egg in the oviduct and hence successful fertilization. During this journey the number of sperms is drastically reduced. Natural sperm selection consists of an array of physical and chemical processes and interactions. Sperm movement properties were examined in various experiments by closely observing sperms *in vitro*. For example sperms swim along surfaces (thigmotaxis) and against fluid flows (rheotaxis). However, it

**Funding:** This work was funded by the Deutsche Forschungsgemeinschaft (DFG, German Research Foundation) under Germany´s Excellence Strategy – The Berlin Mathematics Research Center MATH+ (EXC-2046/1, project ID: 390685689 to EK) and IRTG2290 (to EK). The funders had no role in study design, data collection and analysis, decision to publish, or preparation of the manuscript.

**Competing interests:** The authors have declared that no competing interests exist.

is unclear how those interactions affect fertilization success, as it is not possible to observe sperms after natural mating. Therefore, we present a computational model of sperm migration, which approximates the shape of the female reproductive tract and integrates different interactions between sperms and their surroundings. The model gives insights into the impact of different processes on the number of sperms which reaches the oviducts as well as pointing out which sperms are more likely to succeed. Thus, it gives hints which interactions could be most beneficial for fertilization and hence should be considered in selecting sperms for artificial insemination techniques.

## Introduction

Mammalian reproduction is fundamental to higher life on earth. Yet, our understanding of what creates a successful reproduction event is still lacking detail. This in turn hinders optimization of artificial fertilization efforts in both humans and endangered species. The development of sexes, anisogamy and inner fertilization culminated in the appearance of viviparous birth as a trait of nearly all mammalian species in addition to lactation [1]. In mammals up to billions of ejaculated sperm are deposited in the female vagina, cervix or uterus [2, 3] and have to travel from the site of semen deposition through the entire female genital tract—which is orders of magnitude larger than the sperm itself—to the site of fertilization. Consequently, the genital tract faces the requirement to assist sperm on its way to the female oocyte and to assure that only the fittest succeed in fertilization. Until ultimately one sperm fuses with the female oocyte, a dramatic reduction of sperm number occurs based on stochastic or selective processes. Deep understanding of these processes and the resulting selection gains importance since reproductive success is at risk in humans and in endangered animal species. Between 1973 and 2011 sperm concentration in human semen from unselected men has decreased by 52% (from 99.0 to 47.1 million per ml) [4], approaching the critical sperm concentration considered minimal for native fertilization (20 million sperm per ml) [5]. Therefore, assisted reproduction techniques will support human reproduction under recent social conditions [6] and contribute to animal species conservation. However, bypassing natural selection and conditioning of gametes, as required for *in vitro* fertilization (IVF) techniques, can provoke detrimental consequences for the resulting progeny and the species. Many hypotheses on how spermatozoa are guided and selected on their way to the oocyte have been established [7–12], but most details of the anticipated processes remain undescribed. Analyzing these processes helps to assess the significance of the observed decline of human sperm concentration in industrial countries, to optimize sperm preparation for artificial insemination, or to improve sperm selection for IVF. However, experimental options to investigate sperm transit *in vivo* are limited for ethical and practical reasons, and most information stems from *in vivo* studies in farm animals [13], or from sperm behavior *in vitro*. Primary cell culture models and recent microfluidic 3D cell culture models [14] were also applied to study gamete interactions with different parts of the female genital tract recovered from female animals after slaughter or natural death. To understand complex biological systems, predictive mathematical models are required. Describing the process across multiple levels of biological organization (multi-scale) is vital for comprehensive and predictive modelling of a complex process such as the movement and selection of sperm. Here, we analyze sperm transition in the female reproductive tract integrating hypotheses and experimental knowledge with a spatio-temporal multi-scale biophysical model. We established a three-dimensional reconstruction of the female reproductive tract based on implicit functions. This reconstruction serves as environment for an agent-

based model (ABM) for sperm migration from vagina to oviduct. In ABM, an agent is a freely moving, decision making entity, which can interact with its environment and other agents. ABM allows to accommodate the sperm properties important for sperm migration. Furthermore, potentially relevant factors as sperm mortality, active motion characteristics, and guidance by rheological and geometrical conditions can be assessed. Experiments on sperm transport in domestic animals revealed that out of the vast number of deposited sperm only some hundreds pass the utero-tubal junction (UTJ) [15, 16], the small connection between uterus horn and oviduct, where fertilization occurs. It takes a few hours to accumulate a sufficient number of sperm in the oviduct to ensure fertilization [13, 15]. Our dynamical model for the propagation of sperm to the UTJ predicts key processes, which lead to the reduction in sperm numbers observed *in vivo*. Understanding these processes and shaping constraints of natural fertilization has the potential to greatly improve reproductive success whenever supportive reproduction techniques are required. We first present the mathematical reconstruction of the female genital tract, sperm movement and interaction rules, and environmental conditions affecting sperm migration. The terms sperm and agent are used synonymously. So far, we only considered the fate of sperm in the lower female reproductive tract and agents reaching the UTJ are counted as successful sperm. Second, the model is simulated for different scenarios and hypotheses. The bovine reproductive system is used as example since advanced knowledge is available.

## Materials and methods

An agent based model for sperm propagation was developed using *Python*. It is available on gitlab (https://ford.biologie.hu-berlin.de/jorin/female_sperm_selection). The model initially assigns positions within the cranial vagina compartment to each sperm agent. Furthermore, it sets values for each agent property shown in Table 1. Positions and agent properties are represented by arrays, such that the i-th agent was represented by the i-th position of those arrays. Sperm propagation was modelled using update rules for agent position $\vec{p}_{s,t}$, orientation $\vec{u}_{s,t}$ and speed $v_{s,t}$ in each time step $\Delta t$. Agent movement was restricted to defined shapes, i.e. the reconstructed female genital tract or the box mimicking a specimen chamber. When fluid flow was considered a grid holding fluid velocities was calculated before agent initialization. The spatial resolution of the grid was 20 μm. During simulation agents were exposed to the fluid flow at the nearest point in the grid. Due to the large number of agents, parallelization of the simulation was needed to obtain reasonable simulation times. Parallelization was achieved by usage of a bash script.

### Agent movement

The movement of sperms was described by three rules, (i) by random re-orientation in each timestep, (ii) by alignment along compartment boundaries and (iii) by orientating against the fluid flow. Random re-orientation was implemented as deflection of the agent orientation $\vec{u}_{s,t}$ by the deflection angle $\theta_s$, followed by a matrix based rotation around the original orientation vector. Alignment along the compartment boundaries was achieved by averaging the agent orientation with its approximated projection on the compartment boundary. Averaging the orientation vector with the negative fluid flow vector described positive rheotaxis. Here, the fluid orientation vector was scaled amongst others by the fluid velocity. More details can be found in Supplementary Note B in S1 Text.

**Table 1. Major sperm parameters and their origin.** Speed and lifetime were drawn from normal distributions $\mathcal{N}$. The standard deviation of the deflection angle and the initial orientation were obtained from uniform distributions $\mathcal{U}$.

| Parameter | Distribution | Description | Source |
|---|---|---|---|
| $v_s^{avg}$ | $\mathcal{N}(60; 10)\frac{\mu m}{s}$ | Average velocity | Hyakutake et al. [19] |
| $\tau_{ls}$ | $\mathcal{N}(24; 6)\text{h}$ | Lifetime | estimated |
| $\theta_s^{SD}$ | $\mathcal{U}(1; 119)$ | Standard deviation of deflection angle | Comparison with Tung et al. [20] |
| $\vec{u}_{s,t}$ | $x, y, z \in \mathcal{U}(0; 1)$ | Orientation | |

## Data visualization

Data storage, analysis and visualization was done in *Python*. Export as vtk-format [17] is possible, which can be visualized by *Paraview* [18].

## Results

### Spatial multi-scale reconstruction of the bovine female genital tract

The genital tract is considered as a system of connected tubes with variously folded surfaces (Fig 1 and Supplementary Note A in S1 Text). We divided it into seven distinct compartments, namely: vagina, cranial vagina, cervix, uterine body, uterine horns, UTJs and oviducts [1]. Three-dimensional cylindrical and conical functions were adapted to mimic their shapes. For

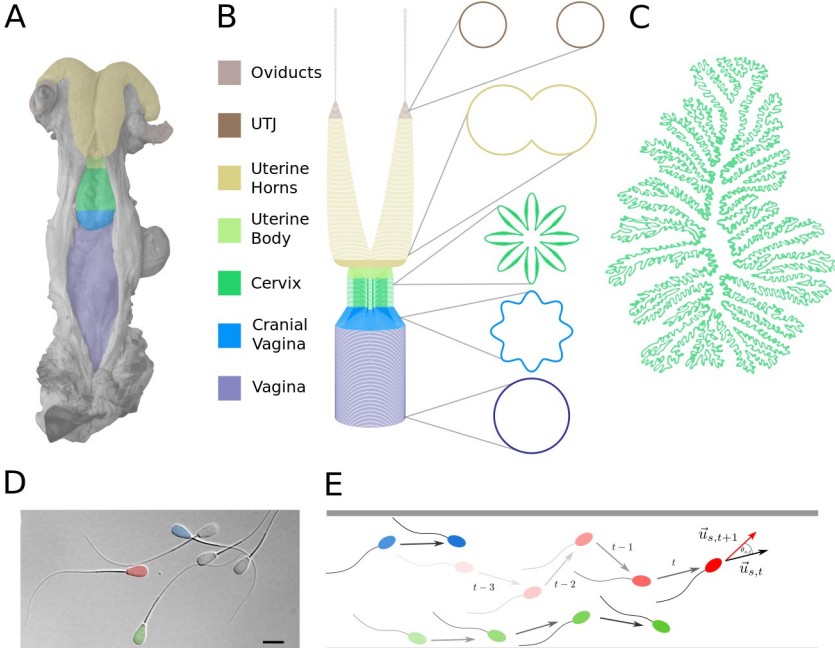

**Fig 1. Mathematical representation of the female genital tract and sperm movement. A** Grey-colored photograph of a bovine female genital tract. Colorcoding indicated separate compartments. The photograph was kindly provided by Dr. A. Peters IFN Schönow. **B** The genital tract is mathematically represented by three-dimensional functions sketched here together with cross-sections at different heights. **C** Sketch of a bovine cervical cross section, adapted from [21]. **D** Differential interference contrast image of bovine sperm, with colorized heads (to highlight individual differences), bar is 10 μm. **E** Sperms are represented as agents with individual properties in ABM, exploring the reconstructed female tract.

instance, the cervix is protruded by primary and secondary longitudinal folds [21], which are mathematically described by trigonometric functions, altering the compartment radius. The individual compartments were combined to an enclosed comprehensive entity by joining them in z-direction. At each compartment transition, e.g. from vagina to cranial vagina, the compartment-describing functions are mathematically identical, ensuring that the system is spatially enclosed.

## Temporal model of sperm movement

In the ABM, each sperm cell is represented by an agent $s$. The model is initialized by assigning a random initial position $\vec{p}_{s,t_0}$ within the cranial vagina to each agent, resembling vaginal deposition [11]. The agents have several individual attributes, namely i) average speed $v_s^{avg}$, ii) lifetime $\tau_{ls}$, iii) orientation $\vec{u}_{s,t_0}$, and iv) standard deviation of the deflection angle $\theta_s^{SD}$ (Table 1 and Fig 2). In each time step $\Delta t$ a deflection angle $\theta_s$ is drawn from a normal distribution around zero with standard deviation $\theta_s^{SD}$ and a current speed $v_{s,t}$ from a normal distribution with mean $v_s^{avg}$ and standard deviation $v_s^{avg}/10$. The orientation is updated, by deflecting the orientation vector $\vec{u}_{s,t}$ by $\theta_s$. Subsequently, the deflected vector is turned around the original orientation by an angle uniformly distributed between $-\pi$ and $\pi$. This results in truly three-dimensional movement within the reconstructed female genital tract. To calculate the new agent position, the orientation is scaled by the current speed $v_{s,t}$:

$$\vec{p}_{s,t+1} = \vec{p}_{s,t} + \vec{u}_{s,t}(\theta_s) \cdot v_{s,t} \cdot \Delta t. \tag{1}$$

Thus, sperm movement without any interaction is a spatially restricted, unbiased persistent (correlated) random walk, i.e. that an agent's orientation depends on its former orientation and has no directional bias (as the mean deflection angle is 0). This is a widespread concept to describe self-propelling particles similar to active Brownian motion and widely used in biological models [22, 23].

## Box model reproduces *in vitro* dynamics of sperm

Using results of elaborated sperm cell tracking techniques [25, 26] and a descriptive set of movement parameters as defined within the Computer Assisted Sperm Analysis (CASA) [27], we simulated agent movement in a box with a height of 20 µm representing a typical specimen chamber (Fig 2) and tested different phenomena as described below. Basic movement and random walk are shown in Fig 2A and 2D.

Sperm align to surfaces and edges [28], a process called **thigmotaxis** [29]. This behavior is realized by alignment of agents to the compartment boundary, by averaging the agent orientation vector $\vec{u}_{s,t}$ with its projection $\vec{n}_\perp$ on the approximated tangent plane (Fig 2B and 2E and Supplementary Note B in S1 Text). The corrected orientation $\vec{u}_{ns,t}$ approaches the compartment wall. Hydrodynamically, sperm cells are pushers [30], pushing fluid onwards and rearwards while replenishing it from the sides [31]. As a consequence, sperm aligned to a surface have a lower probability to change direction than free swimming sperm. This is implemented by diminishing the deflection angle with increasing alignment (Eq S30 in S1 Text). Nosrati et al. [32] measured bull sperm densities in channels of varying width (50 µm, 100 µm and 400 µm). Depending on their distance to channel wall and corners sperms were classified as wall, corner or bulk swimmer. With increasing channel size the percentage of corner swimmers decreased from approximately 80% to around 30%. The presented model predicts an decrease from approximately 60% to around 10% (Fig 2G and 2H). Hydrodynamic properties determining motion and orientation of sperms at surfaces are addressed in more advanced

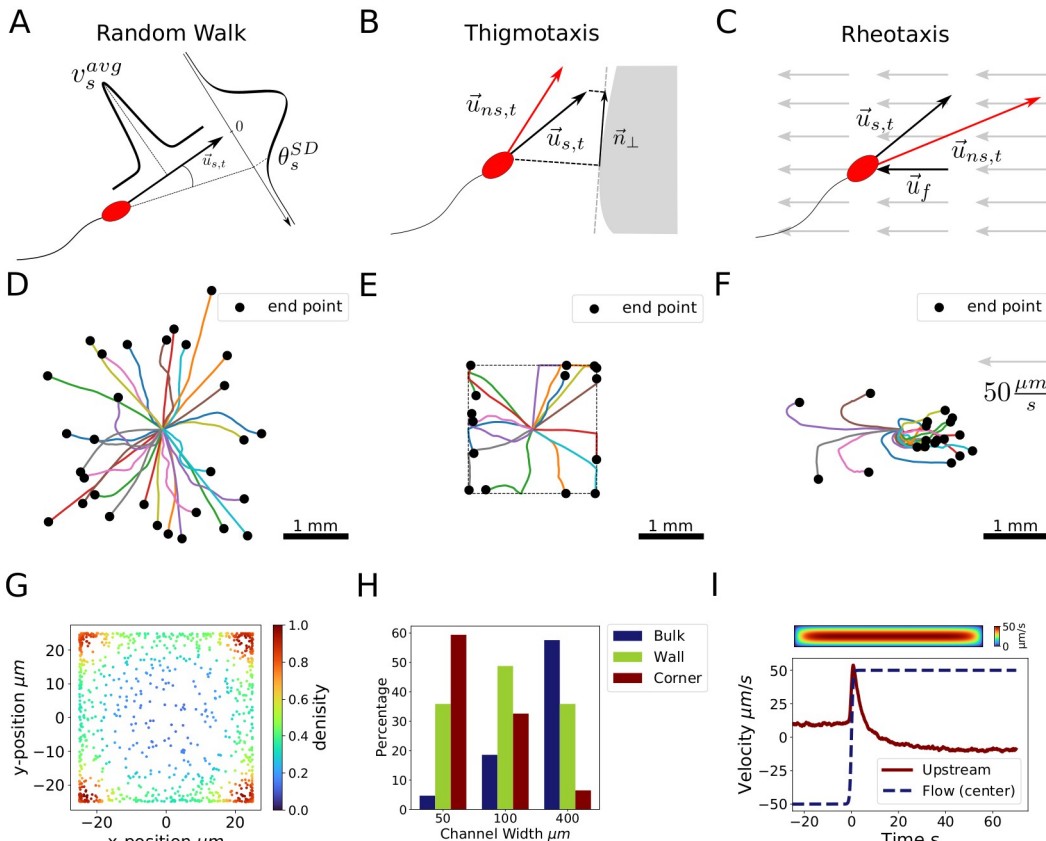

**Fig 2. Agent movement, interaction rules, and resulting trajectories in a box. A** In each time step, agents draw velocities and angles from normal distributions defined by $v_s^{avg}$ and $\theta_s^{SD}$. **B** Thigmotaxis is described by averaging the agents orientation vector $\vec{u}_{s,t}$ with its projection on the tangent plane of the compartment wall $\vec{n}_\perp$, providing the new agent orientation $\vec{u}_{ns,t}$. **C** Positive rheotaxis is implemented by subtracting the fluid orientation vector $\vec{u}_f$ from the sperm orientation $\vec{u}_{s,t}$ (Supplementary Note B in S1 Text). **D-F** Agent movement simulated in a box (120 μm height): agents started in the center, black dots depict final positions. Simulated time was 30 s each. **D** Random walk **E** Thigmotaxis: agent movement restricted by walls (box width and length were set to 2 mm indicated by dashed lines). Agents adapt their orientation when hitting walls. **F** Positive rheotaxis: facing a fluid flow of 50 $\frac{\mu m}{s}$ (direction indicated by arrow), agents reoriented into the flow and moved against it. **G** Cross-sectional distribution of agents ($N = 1000$), simulated in a channel with a width and height of 50 μm. Agent density increases towards boundaries and corners. **H** Depending on their distance to channel boundaries, agents were classified as corner, wall or bulk swimmers. With increasing channel size the percentage of corner swimming sperms decreased, while the fraction of bulk swimming sperms increased. **I** Agents were exposed to a fluid flow reversal in a channel with a rectangular cross-section of 34 μm times 300 μm. The net upstream velocity was analysed for comparison with [24]. After an initial peak on fluid reversal agents re-orientate within 30 s. The inset represents the fluid velocity profile calculated by Eq S41 in S1 Text.

models [30, 33], while our minimal model is based on vector additions when agents are in proximity of the compartment boundary. In contrast to the advanced models our sperms are not attracted towards the wall over long distances. This potentially leads to the slightly lower percentage of agents classified as corner and wall swimmers Fig 2G and 2H, compared with data from Nosrati et al. [32]. As a result, the effect of thigmotaxis is likely underestimated. However, the agents tend to swim along surfaces as shown in Fig 2E.

Mucus is predominantly present in the cervix and potentially guides the sperm towards the uterus and oviducts, as it performs **positive rheotaxis** and orients itself against an oncoming flow [21, 34]. Mucus flow was described by a vector field in which its speed increases

quadratically with distance to the compartment boundary, resembling a Poiseuille profile. The maximum fluid speed $v_f^{max}$ was defined at the lower end of the cervix. Assuming a continuous volume flow through the system, ensuring that no mass vanishes from the system, the mucus velocity $v_f$ at each point in the system was calculated (S6 Fig). The flow direction $\vec{u}_f$ points towards the vagina and is defined compartment-wise (Supplementary Note B in S1 Text). Generally, the faster the fluid and the agent, the better the agent reorients against the fluid direction. Thus, sperm orientation $\vec{u}_{s,t}$ is updated by i) averaging it with the weighted fluid flow direction $\vec{u}_f$ at its position (cross product of $\vec{u}_f$ and $\vec{u}_{s,t}$) and ii) a logistic term, resulting in faster alignment when sperm swims perpendicular to the flow and when fluid velocity is above a threshold, as sperm only align in sufficient flow rates [20] (Fig 2C and 2F and Eqs S47 and S48 in S1 Text). To justify the outcome of our minimal model in comparison with a more detailed model by Kantsler et al. [24], we tested the behavior of our agents upon fluid flow reversal. In agreement with the referred study, after an initial peak in the upstream velocity, agents re-orientate within 20 s (Fig 2I).

## Cell persistence reproduces experimental data

Knowledge is sparse on the angular deflection of sperm per time. Within CASA, straightness (STR) is calculated by the vector length of displacement divided by the contour length of a sperm trajectory. Tung et al. [20] measured STR values of 0.87 ± 0.02 for bovine sperm tracked for 2.81 s. Using this value we estimated the deflection angles for sperm movement, by simulating agent movement in a box and calculating their STR [20, 35]. The agents were positioned in the middle of the chamber, restricting movement to z-direction. Calculation of the STR showed that (i) it is timestep independent, which is ensured by the Euler-Maruyama method [36], and (ii) that it perfectly agrees with the measurements from Tung et al [20] (S9 Fig).

## Simulations in the reconstructed female tract

To investigate the success of sperm in the female tract and assess the impact of thigmotaxis, mucus flow, immune system, and STR, we simulated the adjusted ABM within the reconstructed genital tract model. Fig 3 shows an example of a typical ABM simulation, demonstrating how sperm explore the genital tract over time (see S1 Movie). For this simulation, a mucus flow (maximal mucus velocity in cervix: 50 $\frac{\mu m}{s}$) was applied and agents move through the cervix into the uterus cavity. At 0 hpi, sperm-representing agents are confined to the cranial vagina, where they are initialized. Agents either obtaining negative z-position or having successfully reached the oviducts were removed from the simulation, respectively.

## Survival rate is modulated by immune system

Semen deposition triggers the invasion of neutrophilic granulocytes, which phagocytize sperm in the reproductive tract during the first hours after insemination [7, 37]. While sperms of most mammalian species survive in the oviduct for a maximum of five days after mating [38, 39], our agents in the less sperm-friendly lower part of the female genital tract have a lifetime drawn from a normal distribution around 24 h (Table 1). Immune system activation shortens the agents' lifetimes. The lifetime decrease is described by a sigmoidal function (S8 Fig). Whenever an agent's lifetime drops below zero, indicating the sperm's death, it is removed from the simulation. Mullins and Saacke [21] proposed that sperm is protected from immune cells within the secondary folds (microgrooves). Therefore, being in a microgroove protects agents from lifetime reduction. Thus, the effect of the immune system depends on the agent's position and the time after insemination and, hence, affects agents differently.

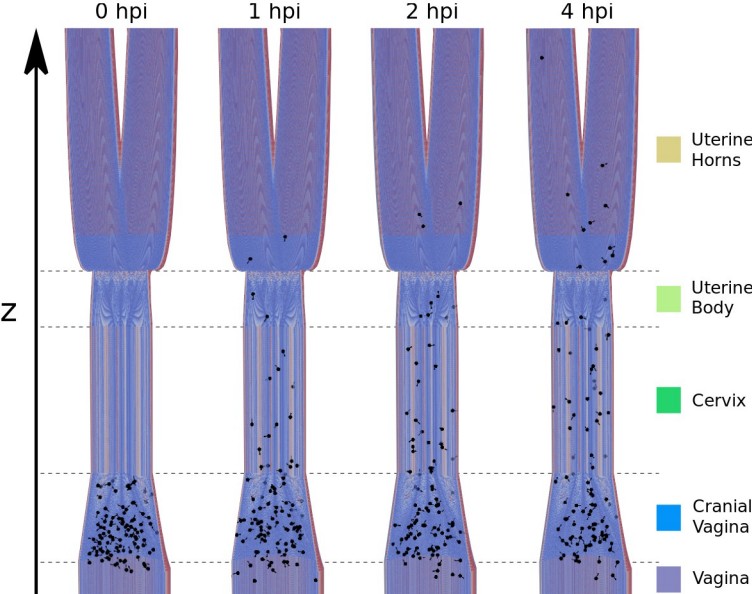

**Fig 3. Sperm migration in the female genital tract.** Agents are positioned in the cranial vagina and explore the space over time in hpi (hours post insemination). The z-position of the agents is the main observable of the system, defining in which compartment an agent is located. For better visibility, sperm were magnified and the genital tract truncated.

## Wall alignment facilitates directed motion

To test the effect of thigmotaxis alone on the agents' performance, we simulated scenarios with and without thigmotaxis (Fig 4). When alignment was omitted, i.e. agents randomly moved through the reproductive tract, none out of 8 million agents reached the oviducts. With thigmotaxis, 336 of 4 million (0.0084%) agents reached the oviducts. Some agents quickly bypass the cervix and move through the uterine horns, while the majority remains within the cervix. After 2 hpi, the first agents reached the oviducts. The agent number drastically decreases between 2.5 and 4 hpi, mostly caused by removal of agents due to the modeled immune system. Around 0.5% of the agent number reduction is caused by agents leaving the system

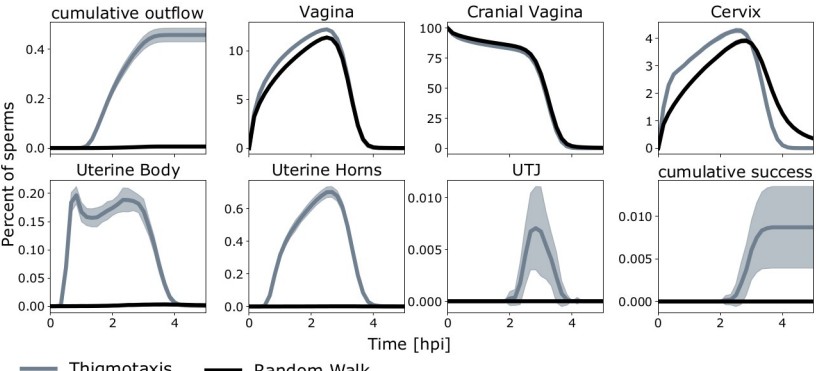

**Fig 4. Agent dynamics in different compartments (in %).** Upper left and lower right: cumulative time courses of agents being removed from the simulation due to negative z-values (outflow) or reaching the oviducts, respectively. Shaded areas indicate standard deviation. Initially, all agents are located in the cranial vagina. Ultimately, almost all agents were removed by the immune system.

through the vagina. To investigate whether the increased success of agents is directly linked to thigmotaxis or if it is a secondary effect due to evasion of the immune system, we simulated long term experiments (48h) without immune system, but with and without thigmotaxis. The results show that even for long simulation times no agent reaches the oviduct without thigmotaxis, while wall alignment boosts success to 0.25%, S11 Fig. Additionally, we simulated agents in a geometry without primary and secondary folds, showing that a wrinkled surface enhanced the effect of thigmotaxis, as the success rate without folds was 0.00545% (S13 Fig).

## Fluid flow aligns sperm motion and boosts their success

To evaluate the impact of positive rheotaxis, we simulated different maximal fluid (mucus) velocities. Agents efficiently align into the fluid flow (Fig 2F). Fig 5A shows the propagation of the agent population through the compartments for different fluid velocities. With positive

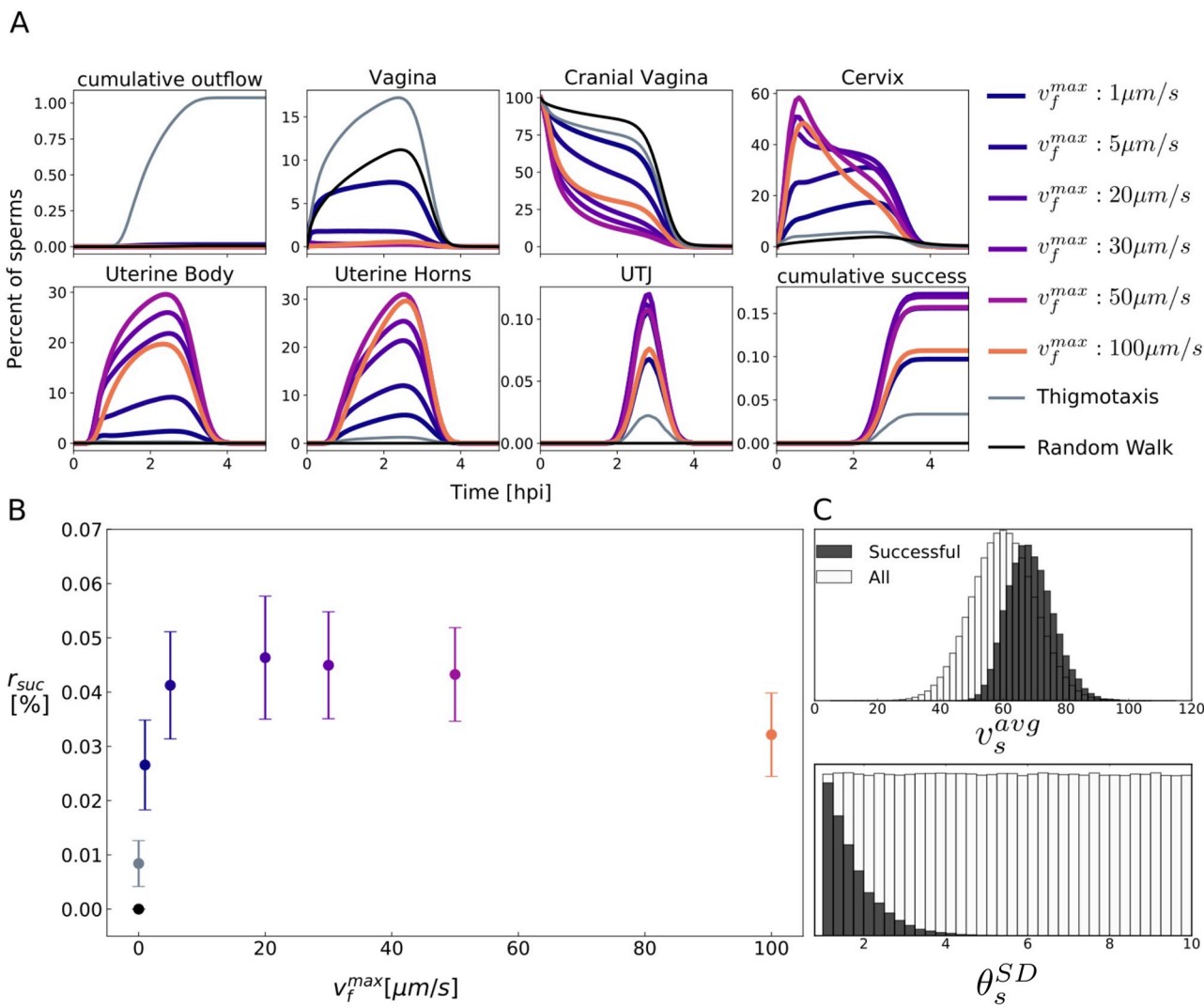

**Fig 5. Effect of mucus flow. A** Percentage of agents per compartment over time. Positive rheotaxis aids agents on their way through the cervix into the uterine body. **B** Success rate of simulated sperm as a function of the maximal fluid velocity. Dots indicate mean percentage of successful sperm, error bars represent the standard deviation. The highest success rate is observed around a maximal fluid speed of $20\frac{\mu m}{s}$. **C** Average speed $v_s^{avg}$ (Successful: $69.9 \pm 7.3\frac{\mu m}{s}$, All: $60 \pm 10\frac{\mu m}{s}$) and standard deviation of deflection angle $\theta_s^{SD}$ of successful and remaining agents.

rheotaxis, in addition to thigmotaxis, sperm are more likely to leave the cranial vagina towards the cervix. Especially for fluid velocities above $20 \frac{\mu m}{s}$, sperm quickly bypass the cervix and begin their transit towards the oviducts. Fig 5B shows the percentage of successful agents as a function of the maximal fluid velocity. The success rate is maximal for medium fluid flows around $20 \frac{\mu m}{s}$. To estimate the maximal percentage of successful sperms we simulated agents in a tract without immune system and a fluid velocity of $20 \frac{\mu m}{s}$ and found that up to 1% of agents are successful without removal by immune responses (S11 Fig).

### Equiangular swimming increases success probability

Next, we investigated, which sperm properties are important for reaching the oviducts, comparing properties of successful agents with the whole population. It appears that faster as well as equiangular swimming agents have a higher probability to arrive in the oviducts, Fig 5C and S12 Fig. Especially linear movement was a major advantage for the successful agents.

## Discussion

Supporting reproduction is an important aim in human medicine and conservation of endangered species. However, an effective application of assisted reproduction techniques requires detailed knowledge of natural processes ensuring reproductive success. During their transit through the consecutive compartments of the female genital tract, sperm are conditioned for fertilization while their number is dramatically reduced. Conditioning and selection events *in vivo* are far from being understood, but extensive experimental tests are mostly impossible—either due to ethical reasons or due to the lack of animals from endangered species. Therefore, we examined a theoretical approach for its suitability to discover patterns and rules of the journey of sperm to fertilization and developed a spatio-temporal model for the mammalian female reproductive tract. Implementing bovine *in vivo* and *in vitro* data, the model incorporates anatomic reconstruction of and a first set of particular sperm interactions with the female reproductive tract in order to investigate mechanisms of sperm selection and propagation. For instance, thigmotaxis and rheotaxis are well known *in vitro* and most likely set a specific environmental condition within the female reproductive tract. Highly sophisticated models are already available to describe the motion behavior of sperm cells in physical environments which mimic native tissue structures [30, 33]. These models are progressively complemented by *in vitro* studies [20, 24, 28, 32]. Since it remains uncertain whether wall alignment occurs *in vivo* and how strong the impact of rheotaxis is, we tested their influence computationally on the success of sperm in our model. Respective simulations of agent behavior in a box were comparable with the above mentioned *in vitro* studies implying a reasonable description of sperm motion characteristics by our rules. In simulations of sperm migration without both thigmotaxis and rheotaxis, no agent reached the oviducts in the given 5-hour-period (Fig 4), after which all agents were dead. *In vivo*, the first few sperm may reach the UTJ and oviducts even within minutes, and a larger functional sperm population is established until 6 to 8 hours after mating [13]. Including thigmotaxis in the simulation leads to a significant amount of successful sperm: By approximating the hydrodynamic properties of sperm, the percentage of agents reaching the oviduct already rose to 0.0084% (0.0055% without primary and secondary folds, S13 Fig). The increase in successful agents in the presence of folds indicates that microgrooves might be beneficial for sperms due to the larger surface area. The aforementioned success rates lie within the range of 0.0001% to 0.1% reported by Eisenbach et al. [11] and Reynaud et al. [2], respectively. However, one should be cautious to compare total numbers for mainly three reasons: First, the immune system was modeled solely as a sperm removing process, which is drastically simplified. Second, idealized functions were used to simulate the

much more complex architechture of gential tract compartments. Third, the deflection angles of sperm were drawn from normal distributions with standard deviations between 1 and 119 degree, which resulted in mean agent straightness values similar to the ones measured by Tung et al. [20]. As the real distribution remains elusive and as a small deflection angle is the main characteristic of sperm reaching oviducts (Fig 5C), a different angle distribution would result in significantly different total number of successful sperm. This refers to one of the model's benefits: to provoke a non-biased re-evaluation of the commonly recorded sperm parameters such as *in vitro* motion characteristics. Further experiments on sperm movement could improve the description of sperm propagation within the model. This also concerns the properties of fluids which surround male gametes under *in vivo* conditions. Considering fluid flow and positive rheotaxis in addition to thigmotaxis could further increase the sperm number reaching oviducts. For a maximal fluid speed of $20 \frac{\mu m}{s}$, the percentage of agents reaching the oviducts increases to 0.046%. Fluid speeds were measured in mouse oviducts at $18 \pm 1.6 \frac{\mu m}{s}$ [40]. Consequently, the fluid velocity yielding the highest success rate in the model lies in a physiological range. However, depending on the fluid velocity a certain volume of fluid has to be expressed in the system. A maximal fluid velocity of $20 \frac{\mu m}{s}$ corresponds to a mucus volume leaving the system of $2.44 \frac{\mu l}{s}$. This would result in a volume of $211 \frac{ml}{day}$. Bovine cervical fluid can reach a volume of $100 \frac{ml}{day}$ [41], which would throughout correspond to the predicted value after simulation. Further, it might be considered that contractions of the female reproductive tract [42] can locally reveal high fluid velocities with smaller volumes of secretions. This could also be valid for the UTJ where the diameter reduction would accelerate the oviduct secretions. In the present model we did not consider a fluid flow from this compartment but it could be asked in future studies whether such a fluid flow could act as physical and/or chemical signal to guide the sperm into the oviduct. In conclusion, a first model with the potential to describe the entire journey of sperm from insemination to passage of the UTJ was presented and provides estimates on the impact of different selection processes. It revealed that physical properties alone would be sufficient to populate the oviduct with sperm and thigmotaxis turned out to be an indispensable process for successful fertilization. Furthermore, it confirmed fluid flow as a major guidance mechanism, skyrocketing the sperm number reaching the oviduct. In general, it is difficult to distinguish between environmental conditions and non-random processes to assess to what extent sperm reduction on the way to the oviducts is due to stochasticity or selection. Therefore, we performed an analysis of successful sperm compared to the total population. Sperm with the highest velocity and persistence within the implemented distributions had a higher probability to reach the oviducts. Whereas the sperm length in its current parameter mode has no significant impact on sperm success (S12 Fig), the length of the principle plus terminal piece of the flagellum in relation to the midpiece seems to be physiologically related to the swimming speed of mammalian sperm as well as the straightness of motion [43]. Therefore, our minimal model would explain experimental data on the observed higher fertility of males with faster sperm. Consequently, the model is helpful to identify sperm parameters, which had evolved by selection pressure and to guide experimental design towards less considered aspects. Sperm successfully undergoing thigmotaxis and rheotaxis should for instance be selected for artificial insemination. In order to answer more detailed questions, more information on genital tract anatomy will be required, e.g. obtaining the exact dimension and geometry from medical imaging techniques or pathological sections. This would open exciting possibilities to simulate sperm transport in rare species where the chance to perform *in vivo* investigations is not given at all. The advantage of the presented model is its extensibility. This concerns for instance a more sophisticated simulation of immune responses. The female immune response develops gradually within the first hours after mating via infiltration

of vagina, cervix, and uterus with leukocytes [7, 37]. Consequently, a progressively increasing reduction of lifetime with time was chosen in the present model as a first simple approximation. However, observations that preferentially viable sperm bind to porcine neutrophils *in vitro* [37] suggest the existence of so far unknown selective processes beyond the removal of damaged sperm. The complex processes occurring to sperm in the oviduct are a further topic to extend/complete the simulation of sperm migration to the oocyte. Chemotaxis, thermotaxis, and rheotaxis as well as sperm interactions with epithelial cells and capacitation-related metabolic changes leading to hyperactivation of sperm motion are currently regarded and investigated as key processes prior to fertilization [7, 34, 44]. Previous models have already focused either on geometry of the oviducts [45] or on the chemotaxis [46, 47]. The molecular conditions in the oviduct are currently under intense investigation by use of cell culture approaches [14]. Exploiting these results will enable the establishment of more realistic model assumptions in the future. Therefore, an extended approach modeling the journey of sperm through all compartments will help to discover patterns and rules for a better understanding of selection processes in the context of species-specific reproductive systems and to optimize assisted reproduction techniques such as artificial insemination.

## Supporting information

**S1 Text. This file contains a detailed description of the model assumptions and equations.** It is structured in 6 Supplementary Notes (A: Reconstruction of the bovine female genital tract, B: Sperm movement, C: Immune system, D: Sperm persistence, E: Thigmotaxis aids transition through cervix and UTJ, F: Computational Execution).
(PDF)

**S1 Table. Parameters used in implicit functions for description of the bovine female genital tract.** Dimensional parameters were taken from Busch et al. [48]. Educated guesses for the number of microgrooves were taken with the help of microscopic images and sketches from Mullins and Saacke [21].
(PDF)

**S2 Table. Sperm population parameters.** Parameters were taken from experimental data [19, 49].
(PDF)

**S3 Table. Individual sperm agent parameters.**
(PDF)

**S4 Table. Temporary sperm agent and other parameters.**
(PDF)

**S1 Fig. Reconstruction of the female bovine genital tract.** Individual compartments (listed on the right) were connected in z-direction. Labelled z-positions indicate compartment transitions and the z-position at which uterine horns divide (Supplementary Note A in S1 Text). Cross-sections at these and intermediate positions are shown on the left.
(TIF)

**S2 Fig. Relative amplitude of secondary fold depth as function of the distance to the primary fold midst.** Dashed purple and green line indicate the beginning and end of a primary fold respectively. Red line indicates the center of the primary fold, while the blue line indicates the relative scaling of the secondary fold depth.
(TIF)

**S3 Fig. Cross-section of the cranial vagina at different heights measured from the vagina.** At 0 cm the cranial vagina equals the shape of the vagina, thus the cross-section is a simple circle. With increasing height first the primary (1 cm and 2 cm) and later also the secondary (3 cm, 4 cm and 5 cm) folds develop. Notice that the secondary folds only occur within the upper half of the cranial vagina (restricted in the condition of Eq S7 in S1 Text by $k_A^{sf}$) and only in the center of the primary folds (restricted by $F_{A_{pf}}$). At 5 cm the cross-section equals the cross-section of the cervix.
(TIF)

**S4 Fig. Wall alignment step by step. A** Sperm orientation is defined by its orientation vector. **B** The solid black vector indicates sperm orientation. Smaller black and blue arrows indicate the scaled vectors, for which it is checked if they lie inside or outside the reproductive tract. The scaling provokes an ellipsoidal shape, indicated by the light grey shaded ellipse. The dashed grey line mimics a compartment wall. The two blue colored arrows, point outside the compartment. **C** The weighted average of the vectors pointing outside (shown in blue) of the compartment defines the normal vector of a plane. Sperm orientation as solid black arrow with label $\vec{u}_{s,t}$. Green arrow indicates resulting normal vector $\vec{n}$. For representational reason it was inverted and enlarged. This normal vector described a plane, shown in dark green. $\vec{n}_\perp$ depicts the projection from $\vec{u}_{s,t}$ onto the plane defined by $\vec{n}$. **D** The new sperm orientation $\vec{u}_{s,t+1}$ vector is shown in red. It results from the sum of the former direction vector and the projection onto the plane. Subsequent the new orientation vector is normalized.
(TIF)

**S5 Fig. Depicted area calculation for two overlapping circles.** $s$ is the distance between the overlapping points. $A_i$ is the green and $A_d$ the orange area, while $A_B$ is the area covered by the radiant $\alpha$. $y_m$ is the distance of one midpoint to the line $s$. Values calculated by Eqs S33-S38 in Supplementary Note B in S1 Text.
(TIF)

**S6 Fig. Flow and velcity profile. A** Volume flow. Red dots indicate heights at which the maximal fluid velocity was set. Blue line corresponds to the continuous volume gain throughout the system. The rate of change is maximal in the middle of the cervix compartment ($z = 34$ cm); **B** Cross-sectional area as function of $z$. **C** Average fluid velocity calculated from continuous volume flux (A) and cross-sectional are (B). **D** Maximal fluid velocity as function of $z$, calculated from the average fluid velocity (C).
(TIF)

**S7 Fig. Sketch on how to determine the center of a conical compartment.** The compartment is depicted by the grey area. $r_l$ and $r_u$ are the lower and upper compartment radii and $l_{com}$ the compartment length. $z_{center}$ is the distance from the cone center to the compartment boundary $z_{offset}$. One has to distinguish between the cases that the upper radius is larger than the lower radius ($\beta > 1$) and vice versa ($\beta < 1$). In the first case, the fluid flow is directed towards the cone center and in the second case away from the center as depicted by the red arrows.
(TIF)

**S8 Fig. Immune system and microgrooves.** Left: Hill function describing the relative activity of the immune system. Right: Agents within microgrooves are protected from the immune system. An agent is defined to be in a microgroove when it is positioned within the modeled genital tract and outside the female genital tract with inverted secondary folds ($A_{sf}^{max} = -0.3$). The figure shows an excerpt of the cervical cross-section of the female genital tract with original

(black) and inverted (red) secondary folds. Shaded areas depict the cross-section of micro-grooves.
(TIF)

**S9 Fig. Agent persistence.** Persistence of simulated agents in a box of 120 μm height. Text above the boxplots gives Mean ± SEM. Orange line indicates median persistence. The persistence shown on the left hand side of the figure originates from a simulation for 2.81 s. For the other persistence the time step was altered and the simulation time was set to 3 s, in order to make persistence comparable between the simulations. Persistence of simulated sperms is in agreement with the persistence reported by Tung et al. [20] (0.87 ± 0.02).
(TIF)

**S10 Fig. Initial alignment beneficial for success.** Mean of alignment score $s_t^{align}$ over time. Successful sperms are compared to all sperms. Especially for simulations with only thigmotaxis ($v_f^{max} = 0$) alignment to the wall within the first 30 minutes (while passing the cervix) increases the possibility to be successful. Independent of $v_f^{max}$ alignment aids the transit through the UTJ (increased mean alignment after 2 hpi.).
(TIF)

**S11 Fig. Percentage of agents per compartment over time in the absence of immune responses.** Thigmotaxis alone helps sperms to reach the oviduct. Under optimal settings of fluid flow up to 1% of sperms surpass the UTJ.
(TIF)

**S12 Fig. Persistent and fast sperms have an advantage.** Pearson correlation between success and sperm properties for different fluid velocities. The speed of sperm correlates positive with success, while a less persistent movement (large deflection angle) correlates negatively with success. Sperm life time shows a small positive correlation with success. Sperm length does not correlate with success in the model.
(TIF)

**S13 Fig. Comparison of simulations with and without folds.** Agents are significantly more successful in geometry with primary and secondary folds. Those simulations were performed with no fluid flow. P-value was calculated with a two-sided t-test from the python scipy package [50]. The Null hypothesis was that both settings have an equal expected value.
(TIF)

**S1 Movie. Propagation of sperms in the female genital tract.** Sperms are positioned in the cranial vagina and begin their journey towards the oviducts (compare with Fig 3). The movie shows sperm migration for the first 4 hours. It results from a simulation with only 50 agents, which were enlarged for better visibility. The maximal fluid velocity $v_f^{max}$ was set to 50 $\frac{\mu m}{s}$. The majority of the sperm remain in the cranial vagina, while others move up the gential tract. The movie was visualized with Paraview [18].
(AVI)

## Acknowledgments

Great thanks go to Dr. A. Peters from the IFN Schönow, who provided the Photograph used for Fig 1.

## Author Contributions

**Conceptualization:** Jorin Diemer, Jens Hahn, Björn Goldenbogen, Karin Müller, Edda Klipp.

**Data curation:** Jorin Diemer, Jens Hahn.

**Formal analysis:** Jorin Diemer, Jens Hahn, Björn Goldenbogen.

**Funding acquisition:** Edda Klipp.

**Investigation:** Jorin Diemer, Jens Hahn, Björn Goldenbogen, Karin Müller, Edda Klipp.

**Methodology:** Jorin Diemer, Jens Hahn, Björn Goldenbogen.

**Project administration:** Karin Müller, Edda Klipp.

**Resources:** Edda Klipp.

**Software:** Jorin Diemer, Jens Hahn, Björn Goldenbogen.

**Supervision:** Karin Müller, Edda Klipp.

**Visualization:** Jorin Diemer, Jens Hahn, Björn Goldenbogen.

**Writing – original draft:** Jorin Diemer, Jens Hahn, Karin Müller.

**Writing – review & editing:** Jorin Diemer, Björn Goldenbogen, Karin Müller, Edda Klipp.

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
