## [Decision Letter · Decision Letter 0]

25 Sep 2020

Dear Prof. Klipp,

Thank you very much for submitting your manuscript "Sperm migration in the genital tract - in silico experiments identify key factors for reproductive success" for consideration at PLOS Computational Biology.

As with all papers reviewed by the journal, your manuscript was reviewed by members of the editorial board and by several independent reviewers. In light of the reviews (below this email), we would like to invite the resubmission of a significantly-revised version that takes into account the reviewers' comments.

Overall, the reviewers appreciate the importance and medical relevance of the model and the main conclusions it produces. However, both reviewers comment on a need to discuss this model in the context of previous modeling work, including its advances and limitations compared to other models. In addition, the reviewers raise some questions regarding the model settings and parameter values that must be addressed.

We cannot make any decision about publication until we have seen the revised manuscript and your response to the reviewers' comments. Your revised manuscript is also likely to be sent to reviewers for further evaluation.

Sincerely,

Stacey Finley, Ph.D.

Associate Editor

PLOS Computational Biology

Jason Papin

Editor-in-Chief

PLOS Computational Biology

Reviewer's Responses to Questions

**Comments to the Authors:**

Reviewer #1: The Reproducibility report has been uploaded as an attachment.

Reviewer #2: This is a high-quality manuscript presenting a comprehensive analysis of sperm migration in a realistic 3D geometry of the female genital tract,

accounting for the most important biophysical processes: thigmotaxis, rheotaxis, immune activity.

The authors report several robust qualitative findings (using bovine fertilization as model system), namely:

- without thigmotaxis, virtually zero sperm reach the fertilization site

- with thigmotaxis, but no rheotaxis, at least 0.01% of deposited sperm would reach the fertilization site

- with thigmotaxis and rheotaxis, this number increases 10-fold to 0.1%

- migration through the reproductive tract selects for faster sperm cells with less directional fluctuations

This manuscript addresses an important topic of both academic and medical relevance,

and should appeal to a broad interdisciplinary readership,

including biologists and clinicians interested in assisted fertilization, and theoretical biological physicists, especially those with an interest in biological fluid dynamics.

The mathematical model is well-crafted and to the best of my knowledge the reported findings are novel.

The manuscript is well-written with a clear structure and exposition of results.

Clear figures explain modeling assumptions in visual form.

The introduction provides a lucid motivation for the study (discussing assisted reproduction and conservation efforts for endangered species).

While the manuscript may be in principle suitable for PLoS Computational Biology,

I want to suggest that the authors address the following comments.

Major comments

1.

I feel the manuscript would strongly benefit if the authors could relate their simple phenomenological descriptions of thigmotaxis and rheotaxis to previously published detailed models.

Ideally, this would provide further justification for the simple phenomenological Ansatzes for thigmotaxis and rheotaxis used e.g. in Eqs. (S28) and (S47).

Similarly, effective parameters in the current model could be derived as "best fits" to previously reported detailed models.

There exists a number of detailed theoretical models for thigmotaxis and rheotaxis of individual swimmers in the literature:

- For thigmotaxis, e.g. both Jens Elgeti & Gerhard Gompper as well as David Smith & Kirkman-Brown published several papers on boundary alignment of swimming sperm cells. In particular, Elgeti et al. compared these detailed models to a minimal model of a self-propelled cylinder interacting with a boundary wall, discussing in detail the confined rotational diffusion of the swimmer's orientation vector [Elgeti et al. EPJ-ST 2016].

- For rheotaxis, Kantsler et al., proposed the effective 'weather-vane model' to explain the anti-alignment of actively swimming sperm cells with external flow [Kantsler, Goldstein et al. eLife, 2014]. While this might have been the first theory paper on rheotaxis after the experimental findings by Miki et al., there are many more theory papers that followed thereafter (which can be found among the 271 citations of Miki et al. as of today)

I urge the authors to compare their minimal models to these previous more detailed models.

This will provide a crucial justification for their minimal models used in their study.

2.

As will most mathematical models of a certain level of detail, this model comprises a large number of parameters.

I appreciate that parameters are presented in tabulated form and references are provided whereever possible.

Nonetheless, a sensitivity analysis would substantially strengthen the reported results.

- What are the key parameters that influence fertilization success most?

- Would a simpler model, say without the undulating surface of the reproductive tract, yield similar results?

- What role does the variability of flagellar length play?

Medium comments

3a.

How would results change for longer search times, or in the absence of immune activity?

3b.

The authors show that thigmotaxis improves fertilization success.

But it is not directly clear if this is a direct consequence of this motility behavior,

or if thigmotaxis helps sperm cells to hide from immune responses inside the grooves of the reproductive tract.

Simulations of thigmotaxis in the absence of immune activity should answer this question.

3c.

How were the parameters for modeling the immune response determined?

4a.

For their rheotaxis model, the authors assume that flows starts below the UTJ compartment,

but that there is no flow in the oviduct itself (page 12, B, bullet point 1).

However, Miki et al. reported flows in the oviducts of mice, and proposed that these induce a relevant rheotaxis response

[Ref. 30: Miki & Clapham, Curr. Biol., 2013].

The authors should discuss this discrepancy.

4b.

For typical flow speeds in the reproductive tract, the authors cite Miki et al., who measured 20um/s in murine oviducts.

However, the radius of the oviduct is much smaller than the radii of the uterine cavity or cervix.

Thus, assuming the same flow speed in these compartments as in the oviduct would actually correspond to a substantially larger total volume flow.

Can the authors justify why the flow speeds considered represent the physiological range?

5.

The authors mention additional biophysical processes in the discussion, but stay very brief on this.

I feel the manuscript would benefit if this section would be extended and references to the relevant literature were added.

- For example, what is the supposed role of progesterone-induced motilty responses?

Will these play a role only in the oviduct, or also in other compartments of the reproductive tract?

Can the authors review the state of the field whether progesterone induces chemokinesis or chemotaxis?

- Sperm cells interact with the ciliated epithelium in the oviduct:

To what extent is this interaction captured by the corrugated fine structure of the oviduct surface with primary and secondary folds as used here?

- The cervix is visco-elastic, while all fluids are treated as Newtonian fluids in this study:

Can the authors predict how the reported results would change qualitatively in a non-Newtonian fluid?

- Is it known at which stage of a sperm cell's progression through the reproduction tract capacitation occurs?

When do sperm cells become hyperactivated?

- Penetration of the cumulus layer represents an additional obstacle for fertilization that should be mentioned at least.

- Thermotaxis is a concept that might need discussion: there seems to be some indirect evidence from the Eisenbach laboratory but no firm confirmation and certainly no consensus in the field.

Even if the present study captures only the migration to the oviduct, and does not yet account for additional mechanisms inside the oviduct,

(which are insufficiently understood at present) , this study represents an important contribution that should be published.

However, as a service to the reader, this manuscript should connect to the current state of the field.

6.

The authors calibrate the 'standard deviation of the deflection angle' in their model,

by comparing simulated trajectories to a previously measured straightness parameter STR.

However, it makes a difference whether such a comparison is done for 2D- or 3D-trajectories.

Is it correct that the previous experimental data by Tung et al. reports on effectively two-dimensional sperm trajectories in a shallow observation chamber

[Ref. 18: Tung, Suarez et al. PNAS 2015]?

If the authors likewise simulated 2D-trajectories whose statistics matched those of the experimental trajectories,

then the 'standard deviation of the deflection angle' should be multiplied by a factor 'sqrt(2)' if I am not mistaken.

I ask the authors to give attention to this technical point.

7.

The authors describe sperm motion as a three-dimensional persistent random walk with fluctuationg speed, which seems an admissable first approximation.

As a service to the reader, the authors should relate this model to similar models such as persistence random walks, Active Brownian Particles, correlated random walks, etc., all of which embody essentially the same idea.

With a little bit of work, I think the authors may even find an analytical formula that links the three analogous concepts

- STR (for a given trajectory length; called 'persistence' in Ref. 18: Tung et al. PNAS 2015)

- 'standard deviation of the deflection angle'

- persistence length

8.

While the model is truely three-dimensional, a casual reader may actually miss this point.

For example, Fig. 2 and Fig. 3 give the visual impression of a 2D model.

Also, the description of persistent sperm motion of 2D reads like the description of a two-dimensional persistent random walk - only in the Methods section does it become clear that the authors consider a three-dimensional persistent random walk.

9.

While discussing Poiseuille flow, the authors write "speed increases quadratically with distance" [introduction & p. 12]

whereas 'u(d) ~ d_max^2 - (d_max-d)^2', see also Eq. (S40).

I find the formulation at least unclear, as it seems to wrongly suggest 'u(d) ~ d^2'.

10a.

I did not immediatedly see why Eq. (S28) ensures time-step independent alignment.

A short derivation or reference would be appreciated.

10b.

For a physicist, an expression like '2^dt' as in Eq. (S28) is not defined because

the time-step 'dt' has physical units of a time and can thus not used as an exponent.

The authors should state clearly how '2^dt' should be interpreted

[even if it is just '2^(dt / 1 second)' because '2^(dt / 1 year)' would be equally fine].

11.

The Ansatz for the microstructure of the compartments Eq. (S2)-(S24) contains many unknown parameters.

The authors should explain in more detail how they achieved their "educated guesses" for these parameters (listed in Table S1).

Could these parameters probably be estimated from microscope images?

12.

I suppose Eq. (S46) contains a typo, because a scalar on the left-hand side is equated to a vector on the right-hand side.

Should one take the norm of 'u_f x u_{s,t}' ?

13.

I would have thought that the length of the sperm flagellum is tightly controlled.

What evidence is there for the assummed variability of flagella length?

(Judging this variability from microscope images can be difficult because the distal tip of flagella often is difficult to track.)

14.

The authors state themselves that their simulation framework is extendable.

Did the authors consider making their simulation code freely available as open source?

15.

For the rheotaxis model given in Eq. (S46), the rate of alignment is proportional to the exteral flow velocity (which is plausible),

but also proportional to the swimming speed of the sperm cell itself, which is less clear.

Can the authors motivate this modeling assumption better?

Minor comments

- The authors may want to consider to add a "z-axis" to Fig. 1B.

- The authors sometimes denote the time step sometimes by 'delta t', sometimes by 'dt' : I assume these are the same?

- Page 3: While introducing the maximal flow velocity '50 um/s', the authors may already quote the measurement from Miki&Clapham, to inform the reader about the expected order of magnitude.

- Figure 2DEF: add scale bar? add label 'end point' at black dots?

- Reference 7: journal missing?

- Page 14: Isn't Eq. (S40) just the same as Eq. (S39) with 'd_max=r_com' and 'd=d_max-r' ?

- I appreciated Fig. S4 but think it could be extended.

For example, to be better able to follow the text on the "ellipsoidal sensing zone" on page 12, an illustration similar to Fig. S4B would be very useful.

- Supplementary Note 2A:

I found the cross-references to the corresponding Figure S4 rather sparse. It would help the reader if in each paragraph reference to the corresponding panel of Fig. S4 would be made.

- References should be provided for the pusher-type swimming of sperm cells.

Reviewer #3: The paper entitled, "Sperm migration in the genital tract - in silico experiments identify key factors for reproductive success" describes an agent-based simulation of sperm migration through the female reproductive tract. Like previous work published in literature, the main focus of the paper is the migratory characteristics of the spermatozoa and the impact on the outcome which is the number of spermatozoa reaching the oviduct.

Generally I like the paper. It shows a simple approach and a very simple strategy to model the sperm migration in the female reproductive tract. Maybe the most important feature of this paper is the inclusion of the potential impact of the immune system in the spermatozoa transport within the female reproductive tract and how it can affect the number of spermatozoa reaching the site of the fertilization. The other interesting aspect of the paper is the author's attempt to include a very simple approach for inclusion of fluidity dynamics in their calculations and in their model.

In my opinion the same issue of simplicity is probably also the main weakness of this paper. The authors have reconstructed the 3D geometry of the tract, but using idealised functions to describe the different cross-sections, rather than medical images. There is no additional representation of internal geometry/folds as far as I can make out. This point needs to be discussed and needs to be mentioned as a limitation of the current paper.

The model incorporates the vagina to the UTJ, but not the oviduct itself. They consider the effects of fluid/mucus flow, but only as a persistent velocity vector that is taken into account when calculating sperm direction, not anything more sophisticated. The sperm behaviour is represented as a simple motion, without consideration of many other aspects of spermatozoa physiology, for example capacitation, hyperactivated motility is included. Also, no consideration of underlying processes in the female reproductive tract e.g. the effect of the reproductive cycle, hormones and even the spatial topography of the reproductive tract in the cow. These points are hardly mentioned in the discussion of the paper. The reader needs to be informed of all the limitations of the paper and it needs extensive discussion.

I think the paper would be improved if authors present a more comprehensive account of other papers in the field and in particular mentioning the other models produced in other species. A more in depth comparison with their approach would be very useful. In addition it will be very important to discuss and mention the shortcomings and limitations of their approach and also the future general avenues for further improvement of modelling spermatozoa migratory behaviour in the female reproductive tract.

**Have all data underlying the figures and results presented in the manuscript been provided?**

Reviewer #1: None

Reviewer #2: Yes

Reviewer #3: Yes

PLOS authors have the option to publish the peer review history of their article (what does this mean?). If published, this will include your full peer review and any attached files.

Reviewer #1: No

Reviewer #2: No

Reviewer #3: No
---

## [Decision Letter · Decision Letter 1]

21 May 2021

Dear Prof. Klipp,

We are pleased to inform you that your manuscript 'Sperm migration in the genital tract - in silico experiments identify key factors for reproductive success' has been provisionally accepted for publication in PLOS Computational Biology.

Please note that the reviewer had two follow-up comments that you may consider in ongoing/future work, but overall is enthusiastic about the revised manuscript.

Best regards,

Stacey Finley, Ph.D.

Associate Editor

PLOS Computational Biology

Jason Papin

Editor-in-Chief

PLOS Computational Biology

Reviewer's Responses to Questions

**Comments to the Authors:**

Reviewer #2: The authors addressed most comments in a deligent manner and included additional analyses to support their results.

I particularly appreciate

- the extended discussion of previous, more detailed models of thigmotaxis & rheotaxis, and a semi-quantitative comparison with their simplified model, as well as

- the availability of the simulation code as open source.

I would thus like to recommend publication in PLoS Computatioal Biology.

For sake of completeness, I want to comment on two responses by the authors, which may still be partially open:

- Comment 4: The flow rates assumed by the authors may still be rather high.

Nonetheless, a plausibility argument has been provided in the revised manuscript.

In the future, precise measurements of fluid flow in the genital tract will hopefully address this gap in knowledge.

- I am still not 100% convinced by the answer to previous comment 15 (i.e., 'rheotaxis alignment rate proportional to sperm swimming speed').

However, given the complexity of the present study, I think it would be acceptable if this point is taken up only in future work by the authors.

**Have the authors made all data and (if applicable) computational code underlying the findings in their manuscript fully available?**

Reviewer #2: Yes

PLOS authors have the option to publish the peer review history of their article (what does this mean?). If published, this will include your full peer review and any attached files.

Reviewer #2: **Yes: **Benjamin M. Friedrich

---

## [Editor Report · Acceptance letter]

18 Jun 2021

PCOMPBIOL-D-20-01294R1 

Sperm migration in the genital tract - in silico experiments identify key factors for reproductive success

Dear Dr Klipp,

I am pleased to inform you that your manuscript has been formally accepted for publication in PLOS Computational Biology. Your manuscript is now with our production department and you will be notified of the publication date in due course.

With kind regards,

Agota Szep
